# Remote Diagnosis on Upper Respiratory Tract Infections Based on a Neural Network with Few Symptom Words—A Feasibility Study

**DOI:** 10.3390/diagnostics14030329

**Published:** 2024-02-02

**Authors:** Chung-Hung Tsai, Kuan-Hung Liu, Da-Chuan Cheng

**Affiliations:** 1Institute of Allied Health Sciences, College of Medicine, National Cheng Kung University, Tainan 701, Taiwan; 070338@tool.caaumed.org.tw; 2Department of Family Medicine, An Nan Hospital, China Medical University, Tainan 709, Taiwan; 3School of Medicine, China Medical University, Taichung 404, Taiwan; u109001022@cmu.edu.tw; 4Department of Biomedical Imaging and Radiological Science, China Medical University, Taichung 404, Taiwan

**Keywords:** natural language, remote diagnosis, GPT-2 model, deep learning, symptom words

## Abstract

This study aims explore the feasibility of using neural network (NNs) and deep learning to diagnose three common respiratory diseases with few symptom words. These three diseases are nasopharyngitis, upper respiratory infection, and bronchitis/bronchiolitis. Through natural language processing, the symptom word vectors are encoded by GPT-2 and classified by the last linear layer of the NN. The experimental results are promising, showing that this model achieves a high performance in predicting all three diseases. They revealed 90% accuracy, which suggests the implications of the developed model, highlighting its potential use in assisting patients’ understanding of their conditions via a remote diagnosis. Unlike previous studies that have focused on extracting various categories of information from medical records, this study directly extracts sequential features from unstructured text data, reducing the effort required for data pre-processing.

## 1. Introduction

Respiratory diseases are common health problems affecting millions of people each year. When individuals come into contact with pathogens like bacteria, viruses, and allergens, their respiratory systems face a wide range of possibilities illnesses. Certain populations with weak immune systems or those exposed to coronaviruses or rhinoviruses might be more susceptible to respiratory problems. The common cold is a convenient term to represent mild upper respiratory diseases, and it has multiple symptoms, like cough, sneezing, sore throat, etc. [1].

To differentiate these respiratory diseases, physicians use a process called diagnostic reasoning [2]. They begin by inquiring about the patient’s complaints, symptoms, and past medical history, which will be collected and translated into a precise and meaningful list that accurately represents the patient’s present condition. Subsequently, physicians correlate these data with their professional knowledge to either affirm or eliminate diagnostic hypotheses. In instances where there is inadequate evidence to support a diagnosis, particularly in hospital settings, additional examinations such as radiological or hematological tests may be conducted to validate the diagnostic hypotheses. By relying on laboratory test evidence, physicians can enhance the accuracy of their diagnosis [3,4]. Patient information and their corresponding diagnoses are recorded in their electronic medical records (EMRs), where diagnostic codes follow the format defined by the International Statistical Classification of Diseases and Related Health Problems (ICD) [5].

The complexities of recorded data are mostly owing to unstructured or non-normalized recordings. Nowadays, patients’ medical records are recorded electronically in digital form such as via the HIS (hospital information system). Most medical systems have unique frameworks and policies on patient privacy. As is stated in [6], there is no standard platform utilized among hospitals in terms of EMRs. In addition, the unstructured information and related medical abbreviations are a challenge for non-medical engineers to utilize and realize these data. This causes difficulty when using the recorded data for research, and the data usually has to be purified.

Remote diagnosis of mild respiratory diseases became necessary during the COVID-19 pandemic era in Taiwan. During the pandemic, all hospitals in Taiwan were controlled and people with non-urgent illnesses were encouraged to stay at home or use a remote diagnosis system such as via the telephone or on-line video consultation. This prevention measure relieves the pressure on hospitals and medical doctors during an epidemic. Also, it diminishes the contact between patients in hospitals. This provided the motivation for this study, raising the following question: Is it possible to create an AI-based diagnosis system, which can be used for people experiencing mild respiratory complications at home or in any place outside hospital to determine a preliminary diagnosis? If yes, this system can be set up as a website or even as an app on any mobile phone.

Compared to text data, quantifiable medical indicators such as biochemical values in blood tests can be analyzed through data mining to identify discriminate thresholds for predicting different diseases [7]. Similarly, image understanding via machine learning or recent deep learning has made a big leap in recent decades [8]. There are thousands of research papers published per year. However, non-quantifiable or unstructured text data has lacked a quantitative analysis approach in the past. A recent rise in natural language processing has led to the era of deep learning in terms of natural language understanding. In the field of natural language processing, language transformation has persistently encountered challenges in achieving effective pattern transformations. Conventional encoding methods, such as one-hot encoding or bag-of-words encoding [9], struggle to accurately capture the features of words and lack of flexibility and/or error tolerance to express different semantics based on contexts. Nevertheless, with the advent of Word2vec technology, we have witnessed rapid progress in the field of natural language understanding. The key innovation of Word2vec lies in its ability to implement dynamic encoding and learning corrections through neural networks. This approach transforms words into high-dimensional vector space representations, effectively addressing the issue of traditional fixed encoding methods being unable to correctly express contextual relationships. The introduction of this technology provides a more powerful and flexible tool for language transformation in natural language processing [10].

Representative techniques in natural language processing models include recurrent neural networks (RNNs) [11,12] and long short-term memory networks (LSTMs) [13,14,15]. These models, based on recurrent processing of textual time series data, excel in capturing information within the context and in understanding semantic structures. Such technologies have laid the foundation for the field of natural language processing, propelling the ability using NNs to understand natural language.

Many algorithms or language models aim to carry out data mining and extract useful properties in unconstructed medical texts. For example, The Unified Medical Language System (UMLS) [16] integrates over 2 million biomedical vocabularies and includes terminology used for bioinformatics. A similar study by the authors in [17] recognizes seven categories from the EMRs based on the language model. In [18], the authors construct a text generation system called MediExpert to assist with differential diagnoses.

However, the language processing models mentioned earlier face the issue of the gradient vanishing when dealing with large-scale time series data. This limitation restricts their performance with long-term data. The phenomenon of gradient vanishing implies that the model struggles to capture long-term dependencies, resulting in inadequate global understanding of the entire time series [19,20]. Recently, researchers have introduced the self-attention mechanism as a replacement for traditional convolution operations. This emerging technology not only enhances the model’s global understanding but also effectively overcomes the problem of gradient vanishing. The self-attention mechanism enables the model to better capture the correlation between different positions in a sequence, leading to significant improvements in handling long textual time series data [21]. The drawback of gradient vanishing makes it difficult to analyze common sequenced sentences.

Figure 1 demonstrates the flowchart outlining our study. In this feasibility study, we collected information from the EMRs of the Department of Family Medicine. In total, 30,592 EMRs were identified and some necessary information was extracted from them such as the ICD, symptoms, and signs. We pre-processed these data and constructed a diagnostic system with a self-attention-based language model, GPT2 [22], to encode our minimal pre-processed data in-house and predict three upper respiratory tract diseases. This diagnostic system can be utilized for validating the efficiency of language models in extracting features from EMRs, serving as a benchmark for verification. This can be further expanded in the future for remote diagnostics.

## 2. Materials and Methods

### 2.1. Dataset Collection and Brief Introduction

The EMRs consisting of patients’ information and physician diagnoses in the ICD format for a total of 30,592 patients were collected from the Department of Family Medicine in Tainan Municipal An-Nan Hospital, Tainan, Taiwan, China Medical University, Taichung, Taiwan from 2017 to 2022. These data were directly extracted from the HIS as comma-separated value (CSV) files and all sensitive personal information was de-identified. Among these data, there are four fields not handled by the system but that are filled in by physicians from patient interviews to final prescriptions. These fields are diagnoses, medical history, medical signs, and treatment. First, physicians record the patient’s narratives, basic physical conditions, and chief complaint in the medical history field. Then, based on previous medical records or further inquiries, chronic illnesses, family medical history, etc., are also recorded in the field of medical history. Furthermore, physicians carry out a physical examination according to the patient’s symptoms and record this result in the medical sign field. Based on the medical history and medical signs, physicians assign at least one and up to three diagnoses for each outpatient using the ICD format for documentation. The field of treatment is completed with the corresponding prescription provided by the physician. Apart from the data in the field of diagnosis, the remaining three fields of the table, namely medical history, medical signs, and treatment, were considered to be unstructured text data, and there are some examples of these text data in Figure 2. This is because for each outpatient, different physicians have different writing styles. However, Taiwanese physicians are not native English speakers, resulting in the records of these three fields containing disorganized descriptions or word fragments. Furthermore, as was found in some records, descriptions were Chinese terms specific to traditional Chinese medicine, having no corresponding English terminology.

The study was conducted in accordance with the Declaration of Helsinki and approved by the Institutional Review Board (IRB) of Tainan Municipal An-Nan Hospital, Tainan, Taiwan, China Medical University, Taichung, Taiwan with the IRB number TMANH112-REC030.

### 2.2. Pre-Processing

Pre-processing consisted of three main steps, namely excluding Chinese and irrelevant data, replacing abbreviations, and filtering out three types of upper respiratory tract diseases based on the ICD classification.

First, it is challenging to translate Chinese descriptions to English automatically using standard medical terminology [23]. Therefore, our data were purified by excluding those records using Chinese descriptions. In addition to this, we observed many clinics conducting remote consultations during the pandemic era of COVID-19. These remote consultations have no relevant symptom descriptions in the medical records and were all recorded in Chinese, so this portion of the data (10,748 patients) was excluded simultaneously. The above exclusion steps were carried out based on the Unicode encoding of Chinese characters. We searched the database in the medical history and medical signs fields. If these fields contained words with the unique encoding associated with Chinese characters, the patient’s data were removed. In total, 16,639 records were excluded.

Secondly, in terms of symptom descriptions, there are many abbreviations and acronyms of specific medical terminology that allow physicians to conveniently record them. However, these abbreviations can pose ambiguity for word embedding, which may also affect the performance of the model. Medical abbreviations are commonly formed by combining the initial letters of words, for example, “F/U” represents “follow up”, and “N/S” signifies “normal saline”. These abbreviations can be challenging for individuals outside of the medical field to comprehend. Moreover, medical abbreviations may exhibit ambiguity, such as the case of “LFT”, which could refer to either “liver function test” or “lung function test” [24]. Accurate interpretation relies on contextual references within the EMR. Therefore, the conversion of abbreviations can alleviate instances of incomprehensibility or ambiguity. Consequently, a manual review of medical records from a six-month period was conducted to select 119 medical abbreviations and establish a dictionary for the conversion of all other textual data,. The purpose of this was to improve the discrimination of the word vector produced from word embedding and decrease the probability of misdiagnosis [25].

Finally, pre-processed data were filtered out based on the ICD classification. Medical records containing the diagnostic codes for nasopharyngitis (460 according to the ICD), upper respiratory infection (465.9 according to the ICD), and bronchitis and bronchiolitis (466 according to the ICD) were selected for this research.

### 2.3. Inputs and Targets

In the database of upper respiratory tract diseases processed with the steps mentioned Section 2.2, we extracted text data from the medical history and medical signs fields as inputs for the diagnostic system. The diagnoses field was utilized to extract ICD codes for subsequent training and testing targets. Providing additional clarification on the targets, for a given patient, there may be up to three diagnosis codes, comprising one primary diagnosis and two secondary diagnoses. These were marked as 1 if they aligned with the focus of our study and 0 otherwise. The target for the diagnostic system was structured with three fields corresponding to the three mentioned upper respiratory tract diseases. If both the primary and secondary diagnoses for a patient fell within the upper respiratory tract disease categories adopted in our experiment, these two diagnoses were concurrently marked as 1, such as ‘[1, 1, 0]’.

### 2.4. Word Embedding

The fundamental unit of a sentence is a word; however, employing individual words as the model input is not a prudent choice. Given the fact that a paragraph may comprise thousands of words, this imposes significant pressure on computer memory. Therefore, it is critical to effectively compress paragraphs. In this study, we chose to tokenize input sentences using the byte-pair encoding (BPE) technique mentioned in GPT-2 [22]. This technique strikes a well-balanced equilibrium between characters and words, preserving representativeness while efficiently compressing the input sentence. For recurrent symbols, the technique merges them into a new token, and this process iterates until the tokenization is complete [26]. Subsequently, each token undergoes transformation based on corpus data, converting it into a unique numerical code. This numerical code serves as the model input, referred to as word embedding, as illustrated in Figure 2. After word embedding, the initial numerical encoding undergoes transformation through the language model (as Figure 2), resulting in a set of 768-dimensional vector features. These features are then utilized for subsequent disease diagnosis.

### 2.5. Language Model

GPT-2, developed by OpenAI in 2018, is a transformer-based language model and an evolution of its predecessor, GPT [21,22]. The self-attention module, a crucial component of the transformer architecture, plays a significant role in GPT-2. It generates query (Q), key (K), and value (V) sets as inputs and applies the following self-attention in Equation (1):(1)$$Attention(Q,\;K,\;V)=softmax(\frac{QK^{T}}{\sqrt{d_{K}}})V,$$
where the self-attention mechanism generates a set comprising query (Q), key (K), and value (V) combinations for each feature input vector [21]; $K^{T}$ is the transpose of *K*; and $\sqrt{d_{K}}$ is a scale. This allows feature vectors from different positions to undergo global attention operations. Such global operations aid in overcoming the locality bias present in CNN models, addressing the issue of gradient vanishing. Consequently, this enhances the model’s capability to handle long-range dependencies among complex features. Self-attention has emerged as a crucial technique for promoting model performance improvement. The complete GPT-2 model consists of 12 blocks incorporating multi-head self-attention, multilayer perception (MLP), and layer normalization, shown in Figure 3 [27]. Multi-head self-attention, as compared to one-head self-attention, allows the generation of multiple sets of global attention mechanisms from the input [21]. Following the extraction of features through 12 blocks, connecting a normalization layer proves beneficial for normalizing the features to facilitate subsequent analysis. GPT-2 closely resembles the original GPT structure, with the exception of the placement of layer normalization and positioning before the self-attention and MLP layers [22]. Notably, GPT-2 models offer numerous advantages over previous language models, like RNN or LSTM, as they mitigate several limitations and allow for the training of larger models in an unsupervised manner [27]. In this study, we employed the GPT-2 model as the backbone to extract features from the tokenized input for our disease classification. This GPT-2 model is constructed based on a model set of Huggingface [28], which is a company providing the platform combining lots of models and applications. We chose the model in the Pytorch version [29].

### 2.6. Classifier

This study turns the disease classification problem to a multi-label classification task [30]. We adopted a multi-task classification approach, treating each disease as an independent sub-task. Specifically, we employed multiple linear layers, with each layer dedicated to classifying a specific disease, shown in Figure 3. The linear layer, namely the fully connected layer, is composed of two hidden layers: the first hidden layer accepts features from the layer norm and performs feature dimension reduction, producing 1024 channels for the next linear layer. The second linear layer accepts features from the previous linear layer and generates two channel outputs for each disease, representing the probability of the existence of that particular disease. This approach possesses several advantages, such as having dedicated classifiers for each disease and facilitating the precise capture of specific features associated with each disease. Additionally, the interpretability of results from each channel is strong, making the model predictions easily understandable. The overall diagnostic model, as illustrated in Figure 3, was implemented using the PyTorch framework.

### 2.7. Training

We adopted the pretrained weights consisting of 117 M parameters released by OpenAI [11] for our GPT-2 encoder. We then conducted fine-tuning of the model using our database and employed cross entropy as our loss function. The cross-entropy loss function was implemented using the ‘’nn.CrossEntropyLoss()’’ module in PyTorch. This module applies the logsoftmax activation function to the input logits and calculates the negative log-likelihood loss with respect to the target values. The ‘’nn.CrossEntropyLoss()’’ formula can be described as:(2)$$Loss=-\log\;\frac{exp\;(x_{n,y_{n}})}{\sum_{c=1}^{C}exp\;(x_{n,c})},$$
where x is the input logic, y is the target, C is the number of classes, and n is the mini-batch dimension [29]. Each linear layer produces an output computed with the target by the cross-entropy function. The three cross-entropy values multiplied by 0.2, 0.4, and 0.4, respectively, were summed as a joint loss function for backpropagation and for updating the weights. Joint loss is described as Equation (3):(3)$$Joint\;loss={loss}_{linear1}\times 0.2+{loss}_{linear2}\times 0.4+{loss}_{linear3}\times 0.4.$$

The weighting factor of 0.2 (mentioned above) was used for nasopharyngitis because this disease represented the largest proportion of our dataset. Therefore, we were keen to lower its weight. Other settings and hyperparameters were as follows: we utilized the Adam optimizer [31], the learning rate was set to 1 × 10^−5^, the max length of input was set to 256, the feature dimensions from the encoder were set to 768, epochs were set to 10, and the batch size was set to 16. This training was executed with one A100 SXM 80 GB HBM.

### 2.8. Evaluation

For evaluation, we employed a confusion matrix to individually analyze the model’s performance for each disease, calculating the values of true positive (TP), false positive (FP), true negative (TN), and false negative (FN) for further evaluation. Additionally, we utilized metrics such as accuracy, sensitivity, specificity, precision, and F1 score, as shown in Equations (4)–(8), to assess the model’s discriminatory performance across various diseases.
(4)$$Accuracy=\frac{TP+TN}{TP+FP+TN+FN},$$
(5)$$Sensitivity=\frac{TP}{TP+FN},$$
(6)$$Specificity=\frac{TN}{TN+FP},$$
(7)$$Precision=\frac{TP}{TP+FP},\;\text{and}$$
(8)$$F1\;\mathrm{score}=2\frac{sensitivity\times precision}{sensitivity+precision}.$$

Accuracy serves as a straightforward metric to assess the model’s overall correctness in diagnosing upper respiratory tract diseases. Sensitivity quantifies the proportion of true positive instances among the actual positive cases, offering an indication of the model’s ability to correctly identify positive instances. Conversely, specificity calculates the proportion of true negative instances among the actual negative cases, assessing the model’s accuracy in correctly identifying negative instances. Both metrics contribute to a comprehensive evaluation of the model. Precision, on the other hand, computes the proportion of true positive instances among those instances diagnosed as positive by the model. Meanwhile, the F1 score represents the harmonic mean of sensitivity and precision, with an ideal value of 1 indicating a scenario where both sensitivity and precision are equal to 1. This metric is particularly useful for evaluating the intersection between sensitivity and precision [32,33]. In addition to evaluation metrics, computational efficiency was also evaluated to identify the trade-off between performance and executing time.

### 2.9. Validation

The overall training and evaluation process was performed using 10-fold cross validation [34]. The ten-fold cross-validation approach involves initially partitioning the dataset into ten equal unique subsets, extracting one subset for testing, and allocating the remaining nine segments to be training and validation sets. The ratio of training, validation, and testing set is 7:2:1. By balancing the differences among the different groups, we can effectively evaluate the model’s performance. In this way, we ensure that the results are not influenced solely by the distribution of the data.

## 3. Results

### 3.1. Screened Dataset

After the data purification described in Section 2.1 and Section 2.2, a total of 20,210 records were collected, and its distribution is shown in Figure 4. There were 7407 cases of nasopharyngitis, 7574 cases of upper respiratory infections, and only 23 cases of bronchitis and bronchiolitis in the single-diagnosis group. However, many patients were diagnosed with not only one case but with two cases. In the two-case diagnosis group, the highest majorities were nasopharyngitis accompanied by bronchitis with 4247 cases, followed by upper respiratory infections accompanied by bronchitis with 956 cases. There were only three instances of nasopharyngitis accompanied by respiratory tract infection. 

### 3.2. Training and Validation

For each patient record, we extracted the “symptoms” field from the text data, representing patient’s current condition, as the input for disease prediction. Figure 5 depicts the loss during the training and validation process. The red and blue curves denote the training and validation loss, respectively. We note that the training loss monotonically decreases to approximately 0.002 in 10 epochs. However, the validation loss shows oscillation in a small range. In order to prevent over-fitting, we stopped the training at 10 epochs to allow error tolerance. We further examined accuracy during the training phase, as shown in Figure 6. In Figure 6A, the prediction accuracy curve for nasopharyngitis steadily increases from an initial 0.87 to 0.99 with respect to epochs, which is almost overlapping with the prediction accuracy curve for upper respiratory tract infection. However, bronchitis and bronchiolitis showed only 0.93 in accuracy. Figure 6B demonstrates the prediction accuracy curves for validation data, which are lower than those for the training data. However, the trend remains consistent with the training data.

### 3.3. Evaluation

To evaluate the model’s performance, we used a test set, showing the prediction accuracies are 0.93, 0.93, and 0.89 for nasopharyngitis, upper respiratory infection, and bronchitis and bronchiolitis, respectively. The sensitivity, specificity, and precision for nasopharyngitis and respiratory infection are above 0.9. For bronchitis and bronchiolitis, the sensitivity is poor at only 0.84. The confusion matrices are shown in Figure 5. The data in the confusion matrix are aggregated based on the number of patients, resulting in an equal sum for all three matrices. As is depicted in Figure 7, the matrices for diagnosing nasopharyngitis and upper respiratory infection exhibit similar trends. Overall evaluation of performance is listed in Table 1; they represent the averages of ten-fold cross validation, which are close to the reality. According to the evaluation result, it is observed that the diagnosis of nasopharyngitis and upper respiratory infection both achieve an accuracy of 0.93, while bronchitis and bronchiolitis reaches a slightly lower accuracy of 0.89. In terms of sensitivity, bronchitis and bronchiolitis exhibit a lower value of 0.84, indicating a higher rate of false positives. A similar pattern is observed in precision and F1 score as well. However, in terms of specificity, bronchitis and bronchiolitis reach a peak of 0.96.

### 3.4. Comparison

The comparative system differs from the originally proposed system in that the encoder portion has been replaced with BERT [35], while the remaining classifier and hyperparameter configurations remain the same. The experimental results, as depicted in Table 2, show that there is no significant difference in diagnostic accuracy between the two systems. In terms of the number of parameters, the BERT-based system has a slightly larger count compared to the proposed method. Regarding FLOPs, the BERT-based system outperforms the proposed method; however, there is no substantial difference in terms of iterations per second between the two.

Additionally, we also studied replacing the purposed multiple classifiers with a sole classifier, which outputted three channels to carry out the multi-label task. The result is shown in Appendix B.

## 4. Discussion

This study explored the feasibility of using few symptom words and a neural network language model (GPT-2) to predict three upper respiratory tract diseases, representing the most common diseases causing patients to seek help from family physicians. Computer-aided diagnosis assists outpatients in rapidly identifying common upper respiratory tract diseases, which might offer information to allow outpatients to seek further medical help. This could be the first step in the remote diagnosis of an upper respiratory tract problem.

Previous studies [17,36] have focused on building annotation systems for electronic medical or health records to extract data such as that regarding symptoms, treatments, and test results. Those data are used for causal inference or defining standard thresholds for subsequent research. However, those applications cannot be directly accessible to the general public. Unlike physicians, the general public has less medical knowledge on the normal ranges of test results but rather focuses on identifying the specific disease they may be suffering from. In this study, we used symptoms recorded by physicians as the input for a language model to infer the disease an outpatient was suffering from. The trained model acts as an artificial diagnostician with extensive experience, helping outpatients understand their conditions.

Compared to the handling of structured text data, this study took a different approach from that of previous research, which extracted various categories of information from the text. Instead, the model directly extracts sequential features from unformatted data to predict diseases. This method reduces the effort required for subsequent data analysis in various categories and eliminates the need for building a guide decision model. Furthermore, the results of this study demonstrate that the model has sufficient capability to determine reliable diagnoses from unstructured text data, indirectly suggesting that physicians can maintain their own writing styles, since the data can be utilized by the language model without requiring adjustments.

Based on the dataset utilized in our study, it was observed that approximately 75% of patients are diagnosed with either nasopharyngitis or an upper respiratory infection. The rationale behind this phenomenon lies in the diagnostic process for nasopharyngitis, wherein physicians can employ visual examination of the throat combined with the patient’s symptom description. For upper respiratory tract infection, even in the absence of overt signs of throat inflammation, the diagnosis is made based on the presence of symptoms related to the upper respiratory tract in the patient’s description. Depending on the severity of the patient’s symptoms, further bacterial culture tests may be necessary for effective treatment. Notably, according to our dataset, patients simultaneously diagnosed with both of the aforementioned conditions constitute a very small minority. Consequently, the distinction in diagnosing these two conditions holds clinical significance, suggesting it is likely that they are not coexisting ailments. Our experimental results demonstrate that our model achieved an accuracy of 93% in diagnosing both of these diseases on the test set. Although there is a slight variance in precision, the results underscore the considerable discriminatory power of our computer-aided diagnostic system. Furthermore, the experimental outcomes indicate that employing different linear classifiers for these two upper respiratory diseases yields satisfactory discriminatory effects, requiring only a minimal number of iterations to achieve a stable diagnostic performance.

In the dataset used in this study, outpatients diagnosed solely with bronchitis or bronchiolitis accounted for only 0.1% of the total, while the data for outpatients diagnosed with these diseases along with nasopharyngitis or upper respiratory tract infections increased to approximately 25% of the total. This data distribution indicates that only a very small portion of patients are independently diagnosed without nasopharyngeal infections or upper respiratory tract infections. In other words, patients with bronchitis or bronchiolitis are usually diagnosed with multiple diseases. Additionally, we infer that physicians generally do not rely solely on symptoms to diagnose bronchitis or bronchiolitis in the absence of chest X-ray evidence.

Regarding the model’s performance, we observe the effectiveness of the GPT-2 model with regard to transfer learning. In terms of the diagnostic accuracy shown in the first epoch of the training process, the accuracy rates for various diseases reach a good level, as shown in Figure 4. This result also implies that the model performs well in terms of few-shot learning, greatly increasing the feasibility of expanding the range of diseases to be diagnosed. After fine-tuning with a relatively small dataset, the transformed learning further improves the performance, especially when acquiring medical data is not trivial. In this study, particularly for the diagnosis of nasopharyngitis and upper respiratory infection, an accuracy rate of 93% was achieved, with sensitivity and specificity both exceeding 90%. Even for diseases with a smaller amount of data, such as bronchitis and bronchiolitis, an accuracy rate of 0.89 was achieved.

Based on our experimental findings, we demonstrate the high credibility of diagnosing using the GPT-2 model, leveraging text vector features generated through the description of symptoms by medical professionals. Each distinct linear layer focuses on examining the relevance within the text features, providing an accurate diagnosis for the specific disease it is tasked with, akin to different specialized physicians. Importantly, these layers are not influenced by biases or preconceptions, allowing for the most precise diagnosis. By employing multiple linear layers for diagnosis, not only does it enhance the interpretability of the classifier, but it also facilitates future expansion to diagnose other diseases. We have the capability to freeze or separate previously trained, fully connected layer weights, enabling the addition of new linear layers without the need for extensive retraining. Furthermore, there is no requirement to adjust output dimensions to accommodate the types of newly added diseases, thereby further enhancing the flexibility of the model. All of these are our contributions.

Since 2020, the outbreak of COVID-19 has led to a rapid increase in the number of patients with upper respiratory symptoms, resulting in insufficient medical capacity. To prevent the outbreak of large-scale infections, the Taiwanese government conducted self-health management policies to request positively diagnosed outpatients to quarantine at home if their symptoms were not serious [37]. The motivation for this study was to provide preliminary help to the general public who suffered with respiratory problems anywhere outside hospitals. In the future, we will extend the language model developed in this study to include the classification of additional respiratory infections such as lung infections.

In our study, a notable limitation is the exclusive use of non-public data from a single center for investigation, leading to significant constraints in terms of the bias of the research results. In future investigations, we aim to validate our model with external data, extending its application to a broader spectrum of diseases by introducing additional classifiers or integrating our trained diagnostic system with other domains. In addition, this study verifies the precision of diagnosing upper respiratory tract diseases using a diagnostic system built on self-attention-based language models. For future research, exploring the integration of voice communication software, like LINE application, LINE Plus Corporation (Yotsuya Office, Yotsuya, Shinjuku-ku, Tokyo, Japan) to allow users to input data could enhance the system’s accessibility and practicality, providing users with accurate preliminary diagnoses in real time.

## 5. Conclusions

We adopted and modified a GPT-based language model applying unstructured medical text data to classify three common respiratory diseases. This method successfully differentiates different diseases from the symptoms recorded by physicians. The resultant performance suggests that this model has capabilities of dealing with complicated text data through NLP. Currently this is only a feasibility study, which is not mature for clinical usage. We demonstrate the possibility that this method has the potential to be used in remote diagnosis.

## Figures and Tables

**Figure 1 diagnostics-14-00329-f001:**
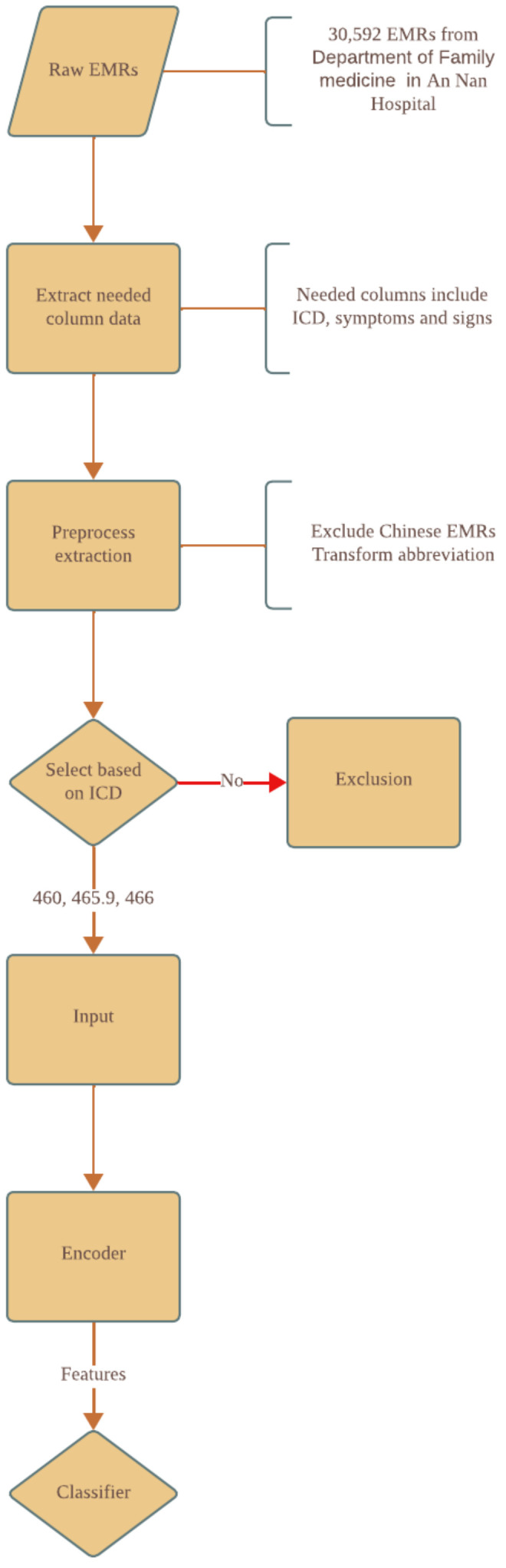
Flowchart of this study. Details of this flowchart are described in Section 2.

**Figure 2 diagnostics-14-00329-f002:**
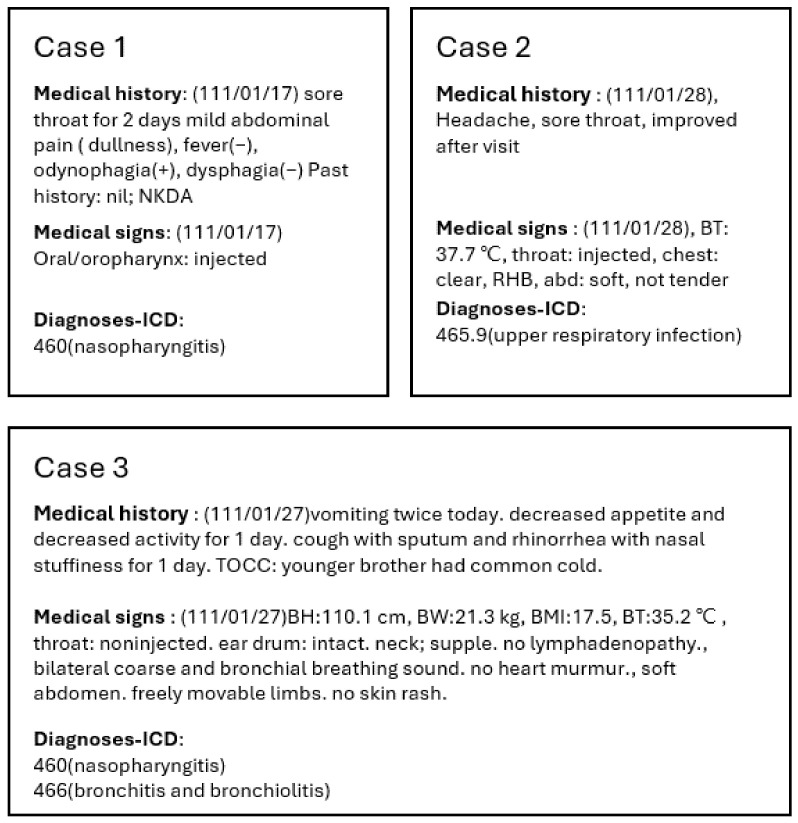
Samples of the dataset.

**Figure 3 diagnostics-14-00329-f003:**
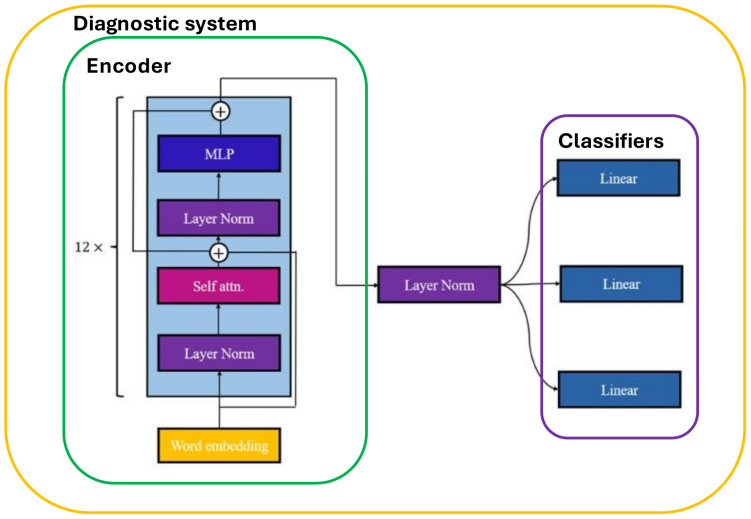
Diagnostic system architecture, which is able to classify three diseases.

**Figure 4 diagnostics-14-00329-f004:**
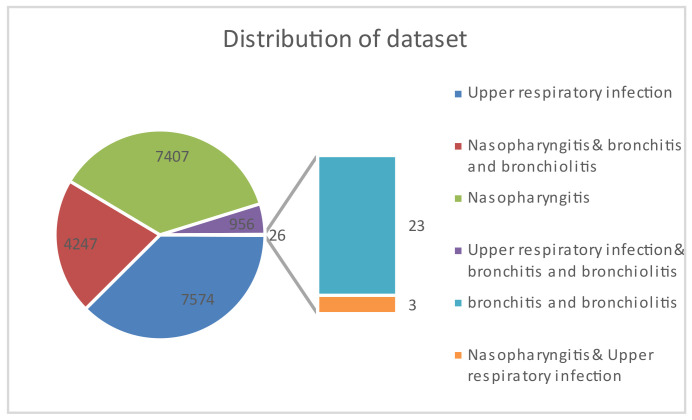
The data distribution; of a single-diagnosis and two-case diagnosis.

**Figure 5 diagnostics-14-00329-f005:**
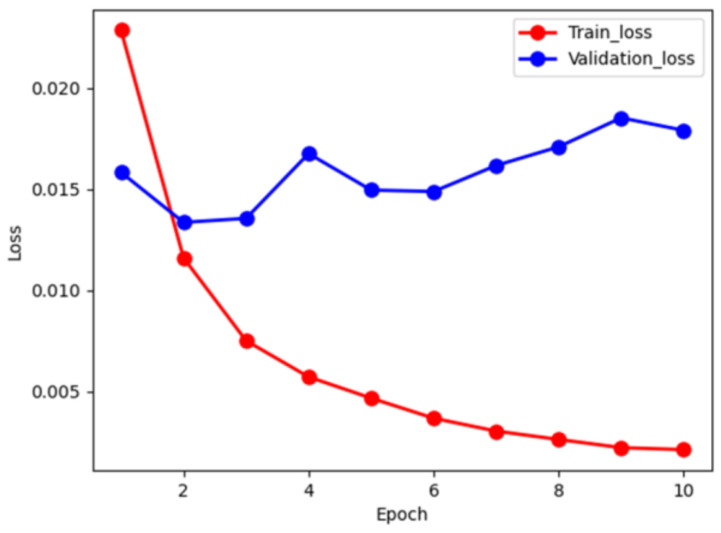
The red and blue curves denote the loss of training and validation with respect to epochs, respectively. The validation loss seems to oscillate in a small range during the first 10 epochs.

**Figure 6 diagnostics-14-00329-f006:**
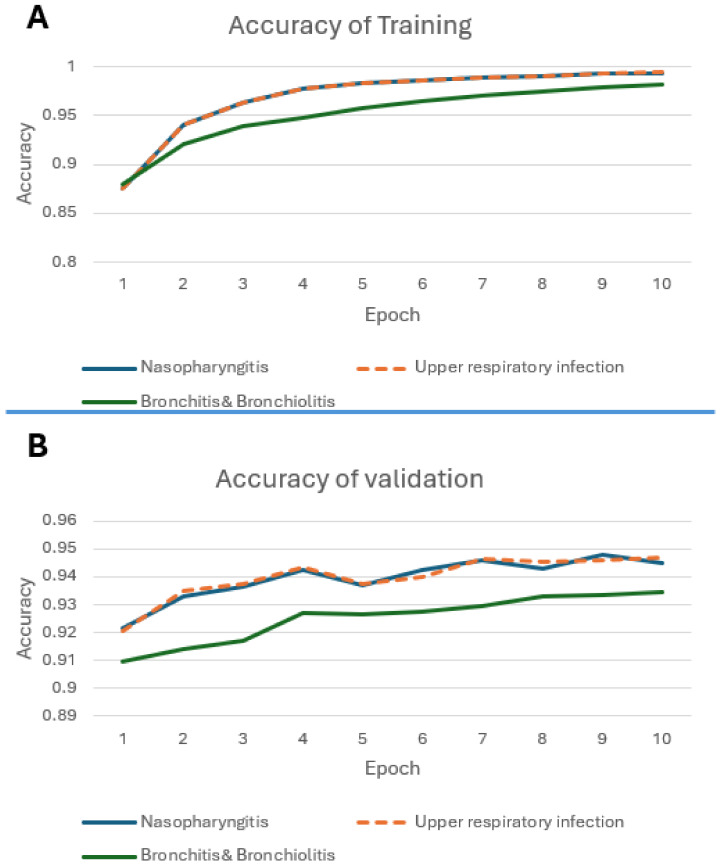
Accuracy curves in training phase. (**A**) Training; (**B**) validation.

**Figure 7 diagnostics-14-00329-f007:**
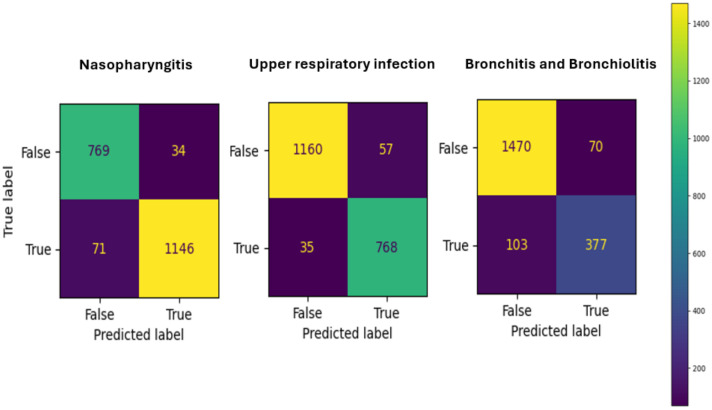
The confusion matrices of three disease predictions.

**Table 1 diagnostics-14-00329-t001:** The evaluation of the trained model. The value is the average of 10-fold cross validation.

Disease	Accuracy	Sensitivity	Specificity	Precision	F1 Score
Nasopharyngitis	0.93	0.93	0.92	0.94	0.94
Upper respiratory infection	0.93	0.92	0.94	0.92	0.92
Bronchitis and bronchiolitis	0.89	0.84	0.96	0.88	0.86

**Table 2 diagnostics-14-00329-t002:** Comparison with other SOTA model. The value is the average of 10-fold cross validation.

Method	Accuracy	Parameter	Flops	Iteration
GPT-2-based	0.93	604 M	1 B	4.87 it/s
BERT-based	0.93	690 M	35.8 B	5.15 it/s

## Data Availability

Data are contained within the article.

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
