# Peer review of "Remote Diagnosis on Upper Respiratory Tract Infections Based on a Neural Network with Few Symptom Words—A Feasibility Study"

_diagnostics, 2024, doi:10.3390/diagnostics14030329_

Round 1

Reviewer 1 Report

Comments and Suggestions for Authors

The study is supposed to present a feasibility study on remote diagnosis on upper respiratory tract infections based on neural network with few symptom words. Accordingly, I assigned several points that would improve the paper:

For the Material and Methods section:

1.     Flowchart flowchart is essential to be included in this section.

2.     There is a lack of description of the neural network model, which should include the following main points:

·       Model architecture: Describe the specific architecture of the neural network used for Upper respiratory tract infection (URTI) diagnosis. This should include details about the number of layers, neuron types, activation functions, and loss function.

·       Model training parameters: Specify the hyperparameters used for training the neural network, such as learning rate, optimizer, and training epochs.

·       Model evaluation: Describe the metrics used to evaluate the performance of the model. This may include accuracy, precision, recall, F1-score, and confusion matrix.

·       Cross-validation: Explain how the model's performance is validated to ensuregeneralizability. This may involve using k-fold cross-validation or a separate test dataset.

In Results and Discussion sections, the study should also include the following discussions:

1. Symptom importance, where the study should explore the relative importance of different symptom words in the model's predictions. This can be achieved through techniques like saliency maps and layer-wise relevance propagation, providing insights into which symptoms are most informative for diagnosis. Identifying the most important symptom words can be valuable for clinicians, allowing them to prioritize questions during patient consultations and improve the accuracy of remote diagnosis.

2. Feature Engineering Techniques, where the study should discuss the effectiveness of different feature engineering techniques for representing symptom words as input to the neural network. This may include techniques like word embedding and bag-of-words representation. Comparing the performance of different feature engineering techniques can help determine the optimal approach for extracting relevant information from symptom words for URTI diagnosis.

3. The study should evaluate the computational efficiency of the proposed approach. This includes measuring the training time and inference time of the neural network model.

4. The study should compare some key factors like the accuracy, generalizability, and computational efficiency of the proposed neural network approach with other existing methods, such as rule-based systems and other machine learning models. This comparison will highlight the advantages and limitations of the proposed approach, providing valuable insights into its potential applications in clinical practice.

·       The study should also acknowledge the limitations of the proposed approach, such as the need for further validation on larger datasets and in real-world clinical settings.

·       Potential future research directions should be discussed, such as exploring the use of additional data sources like medical history and environmental factors for improved diagnosis.

Comments on the Quality of English Language

 Minor editing of English language required

Reviewer 2 Report

Comments and Suggestions for Authors

My comments are the following:

1. The introduction section should include the structure of the paper in the last paragraph. In addition, it should focus on the contribution of this research work.

2. How the dataset (please correct the title in subsection 2.1) was collected? Give some more details about this task. 

3. The Training and Evaluation section should be explained in a better way. The procedure of model building should be analyzed more thoroughly.

4. Sure, the performance metrics for the three diseases are remarkable. But, what is the superiority of the proposed trained model compared to other models?

5. You can add some related references to improve the quality of the paper:

6. Some potential future directions could be added at the end of the paper.

7. Please double-check the whole paper for grammar and syntax errors.

Comments on the Quality of English Language

Some grammar and syntax errors should be corrected.

Reviewer 3 Report

Comments and Suggestions for Authors

The manuscript titled "Remote diagnosis on upper respiratory tract infections based on neural network with few symptom words - A feasibility study" presents a novel and significant contribution to the field of remote medical diagnostics using neural networks. The authors have successfully demonstrated the feasibility of using a neural network model to diagnose upper respiratory tract infections based on limited symptom data, which is particularly relevant in the current healthcare context.

While the study is well-conducted and the results are promising, I have a few suggestions that could further strengthen the paper:

1. Data Representation and Preprocessing: The paper would benefit from a more detailed explanation of the data preprocessing steps. Clarifying how unstructured text data was managed and the rationale behind chosen preprocessing techniques would enhance the paper's methodological transparency.

2. Model Validation and Generalizability: Additional information on the validation process, specifically regarding the diversity and representativeness of the validation dataset, would help in assessing the model's applicability in real-world scenarios.

3. Comparative Analysis: A comparison with existing models or traditional diagnostic methods could provide valuable context. This would aid readers in understanding the relative strengths and weaknesses of the proposed model.

4. Discussion on Limitations: An expanded discussion on the limitations of the current study, including potential dataset biases and the model's performance in complex cases, would provide a more balanced understanding of the research.

5. Future Work and Scalability: Insights into potential future developments and scalability of the model, including how it might be adapted to diagnose a broader range of respiratory conditions, would be beneficial.

Overall, the manuscript is well-written and presents an innovative approach to remote diagnosis. With the suggested improvements, I believe it would make a significant contribution to the field. I recommend accepting this manuscript for publication after minor revisions.

Round 2

Reviewer 1 Report

Comments and Suggestions for Authors

Thanks for considering all my comments